# EXPERT HEADS: ROBUST EVIDENCE IDENTIFICATION FOR LARGE LANGUAGE MODELS

**Qi Wu**
School of Computer Science and Technology
Soochow University
Suzhou, China
wuqi7137@gmail.com

**Jianfeng Qu** [*]
School of Smart Governance
Renmin University of China
Suzhou, China
Suzhou Key Lab of Multi-modal Data
Fusion and Intelligent Healthcare
Suzhou City University
Suzhou, China
jfqu@ruc.edu.cn

**Ximing Li**
College of Computer Science and Technology
Jilin University
Jilin, China
RIKEN Center for Advanced Intelligence Project
Japan
liximing86@gmail.com

**Zhixu Li**
School of Smart Governance
Renmin University of China
Suzhou, China
School of Information
Renmin University of China
Beijing, China
zhixuli@ruc.edu.cn

## ABSTRACT

Large language models (LLMs) exhibit strong abilities in multi-document reasoning, yet their evidence identification is highly sensitive to input order. We trace this limitation to attention mechanisms, where many heads overemphasize sequence boundaries and neglect central content. We systematically analyze attention distributions under document permutations and discover a small subset of heads that consistently prioritize task-relevant documents regardless of position. We formalize these as **Expert Heads**, identified via activation frequency and stability across permutations. Experiments on LLaMA, Mistral, and Qwen reveal architecture-specific patterns: mid-layer heads in LLaMA and Mistral dominate semantic integration, while deeper-layer heads in Qwen specialize in evidence selection. Moreover, Expert Heads exhibit concentrated focus during understanding and more distributed engagement during generation. Their activation strongly correlates with answer correctness, providing diagnostic signals for hallucination detection. Leveraging Expert Heads for document voting significantly improves retrieval and ranking on HotpotQA, 2WikiMultiHopQA, and MuSiQue, outperforming dense retrievers and LLM-based ranking with minimal overhead. Ablations confirm that even a small subset achieves robust gains. Our findings establish Expert Heads as a stable and interpretable mechanism for evidence integration, offering new directions for context pruning, hallucination mitigation, and head-guided training of LLMs.[1]

## 1 INTRODUCTION

Large language models (LLMs) exhibit strong capabilities in aggregating information across multiple documents (Lewis et al., 2020; Wei et al., 2022). Yet, their ability to identify task-relevant evidence is highly sensitive to input order (Liu et al., 2023; Zheng et al., 2023; Pezeshkpour & Hr-

---

[*]Corresponding author

[1]Our code, dataset, and the models used in this work are available at https://github.com/Xuan-Van/ExpertHead

uschka, 2023). This positional sensitivity largely stems from attention mechanisms: many heads overemphasize sequence boundaries and fail to consistently capture critical content in the middle of the context (Press et al., 2021; Guo et al., 2024; Wu et al., 2025). Addressing this limitation requires a deeper understanding of how attention distributes across documents and which components are truly responsible for robust evidence integration.

To this end, we systematically examine LLM attention under context permutations. As shown in Fig. 1, in each model, a small set of attention heads consistently attend to gold documents regardless of their position. These heads appear to play a unique role in identifying task-critical evidence and exhibit resilience to positional variations. This raises an important question: *can we reliably identify such heads and leverage them to enhance both robustness and interpretability?*

We build on this observation by introducing a framework to isolate these heads. We define **Activated Heads** as those that allocate more attention to all gold documents than to distractors, thereby avoiding the pitfalls of naive top-attention selection. We then quantify their stability across permutations and designate the most reliable and focused subset as **Expert Heads**.

Our analysis reveals deeper insights into LLM internal organization. Expert Heads follow distinct layer-wise distributions across architectures: in LLaMA and Mistral, mid-layer heads dominate semantic integration, while in Qwen, deeper-layer heads specialize in evidence selection, highlighting architecture-specific alignment strategies. Moreover, under *Query-as-Source* and *Response-as-Source* attention, more heads participate during answer generation, and their focus is slightly more dispersed, indicating functional shifts across decoding stages.

Through extensive experiments, we show that Expert Heads are closely linked to model performance. When the model produces correct answers, these heads activate more frequently and focus on task-relevant evidence. Conversely, when the model makes errors, their activation weakens or spreads out, leading to insufficient evidence integration and occasional hallucinations. Leveraging Expert Heads for document voting consistently improves identification and ranking across multiple benchmarks (Yang et al., 2018; Ho et al., 2020; Trivedi et al., 2022), outperforming standard retrieval pipelines and direct LLM-based ranking.

In summary, Expert Heads provide a stable, interpretable, and efficient mechanism for evidence integration, revealing LLM internal reasoning patterns, layer-specific roles, and practical utility for improving document identification and ranking.

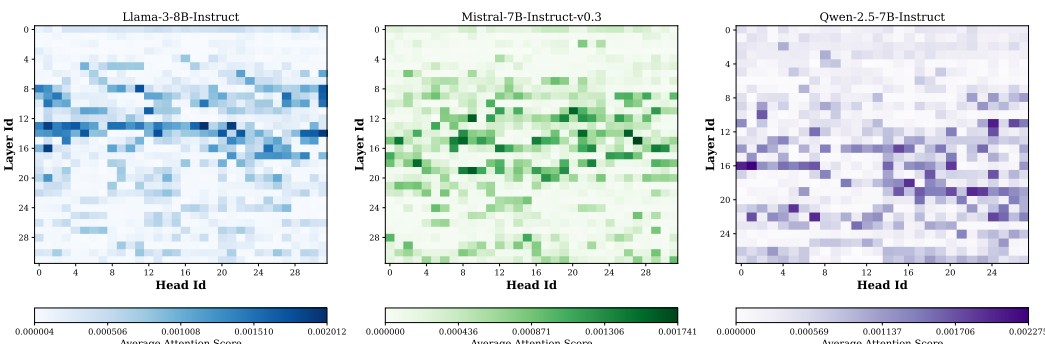

Figure 1: Head-wise distribution of average attention scores for gold documents across three LLMs. Scores are computed by averaging over input permutations where gold documents appear in different positions. Each subplot presents all heads across layers, with color intensity representing the average score assigned to gold documents. The heatmaps highlight a small subset of heads that consistently allocate strong attention to gold documents, indicating their potential role as Expert Heads. Distinct color maps are used for clarity across models.

## 2 ATTENTION HEAD ACTIVATION

To investigate how large language models (LLMs) dynamically allocate attention to task-relevant evidence during context understanding and answer generation, this section analyzes the activation patterns of attention heads under different document permutations.

### 2.1 PRELIMINARIES

**Input Construction and Permutations.** Given a query $Q$, a set of $m$ distractor documents $\{D_1, D_2, \ldots, D_m\}$, and $n$ gold documents $\{G_1, G_2, \ldots, G_n\}$, we construct multiple input permutations to systematically evaluate the effect of document order on attention head behavior. Each input sequence consists of a task instruction, all distractor documents, and the query, with gold documents inserted at different positions among the distractors. Instead of enumerating all possible permutations, we only vary the insertion positions of gold documents relative to distractors. This results in $m + 1$ permutations per query, ensuring that every gold document appears in all possible positions within the context.

**Attention Sources.** For any document $D \in \{G_1, \cdots, G_n, D_1, \cdots, D_m\}$, We consider two types of attention sources:

*Query-as-Source*: Attention from query tokens $Q$ to document $D$, reflecting the model's estimation of relevance during context understanding:

$$A_{Q \to D}^{(l,h)} = \frac{1}{|Q| \cdot |D|} \sum_{q \in Q} \sum_{d \in D} A_{q,d}^{(l,h)}. \tag{1}$$

*Response-as-Source*: Attention from generated response tokens $R$ to document $D$, capturing the evidence actually utilized during answer generation:

$$A_{R \to D}^{(l,h)} = \frac{1}{|R| \cdot |D|} \sum_{r \in R} \sum_{d \in D} A_{r,d}^{(l,h)}. \tag{2}$$

Together, these two perspectives provide a comprehensive view: Query-as-Source indicates which content the model deems important, while Response-as-Source reveals how that content is subsequently used in generation.

**Activated Heads.** For each permutation $\pi$ and attention source $src \in Q, R$, an attention head $(l, h)$ is considered *activated* if it attends more strongly to all gold documents than to any distractor:

$$\text{Activated}(l,h)_{src}^{\pi} = \begin{cases} 1, & \text{if } A_{src \to G_j}^{(l,h)} > A_{src \to D_i}^{(l,h)}, \ \forall j \in \{1, \ldots, n\}, \forall i \in \{1, \ldots, m\}, \\ 0, & \text{otherwise.} \end{cases} \tag{3}$$

This stricter criterion avoids biases from naive top-attention selection, where some heads might appear highly activated due to positional effects rather than true relevance to the task.

### 2.2 ATTENTION ANALYSIS SETUP

**Models.** We conduct experiments on three widely used instruction-tuned LLMs: LLaMA-3-8B-Instruct (Patterson et al., 2022) (32 layers, 32 attention heads per layer), Mistral-7B-Instruct-v0.3 (AI, 2024) (32 layers, 32 heads per layer), and Qwen-2.5-7B-Instruct (Yang et al., 2024; Team, 2024) (28 layers, 28 heads per layer). These models offer sufficiently rich attention structures to enable fine-grained head-level interpretability.

**Dataset.** For reproducibility, we randomly sample 5,000 instances from the HotpotQA (Yang et al., 2018) train set, each containing two gold and eight distractor documents, yielding nine permutations per query. This results in **45,000 input instances per model**, providing a comprehensive basis for analyzing attention behaviors under systematically varied document positions.

**Attention Extraction.** For each input instance, we extract attention maps from all heads across all layers. Both Query-as-Source and Response-as-Source attention matrices are computed using Eqs. 1 and 2. The binary activation status of each head is then determined using Eq. 3. This setup enables head-wise analysis of how attention distribution depends on document ordering.

## 2.3 ACTIVATION PATTERNS OF ATTENTION HEADS

Figure 2 shows that gold documents placed at boundary positions (beginning or end of the context) trigger a larger number of activated heads, but with relatively weaker attention scores. In contrast, gold documents located in the middle elicit fewer activated heads, yet with stronger and more focused attention.

This positional effect is also reflected in performance: as shown in Appendix Fig. 7, multi-hop QA accuracy is highest when gold documents appear at the start or end, but drops when they occur in the middle. This demonstrates that both attention allocation and downstream performance are sensitive to document position.

We further observe notable differences between attention sources. With Query-as-Source, fewer heads are activated, but their average scores are higher, suggesting that a small set of heads suffices to capture task-critical semantics. With Response-as-Source, more heads are engaged, but their attention is more dispersed, indicating a broader integration of evidence during answer generation.

Overall, these findings highlight that LLMs dynamically adjust their attention allocation strategies depending on the stage of processing and the position of critical evidence within the context.

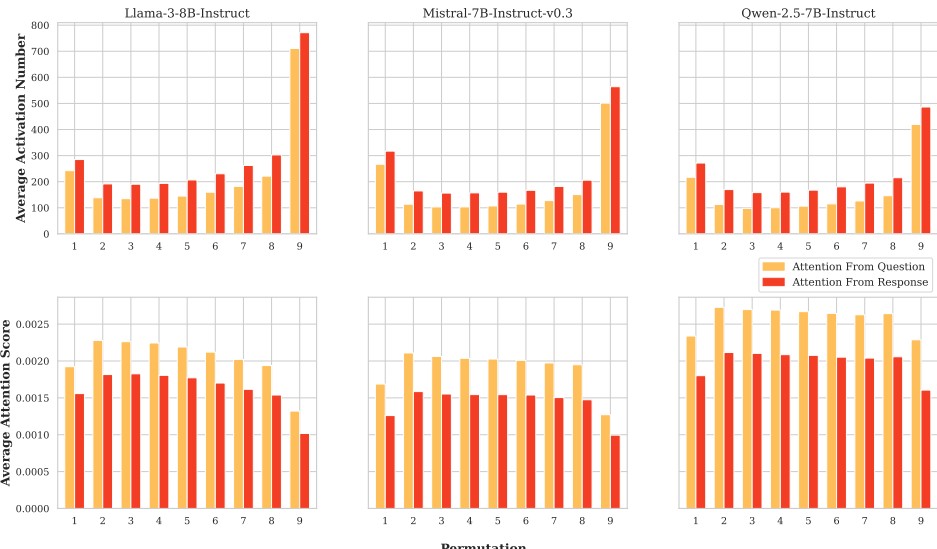

Figure 2: Number of activated heads (top) and average attention scores (bottom) across different gold document permutations. Gold documents at boundary positions activate more heads but with weaker attention, whereas middle positions activate fewer heads but with stronger, more focused attention. The x-axis (1–9) denotes the gold document position within the context, from start (1) to end (9).

## 3 EXPERT HEAD IDENTIFICATION

Building on the activation patterns observed in Sec. 2, this section analyzes the behavior of activated heads to identify those that consistently demonstrate stability and strong focus on gold documents across different document permutations.

### 3.1 DEFINITION

To quantify the reliability and importance of individual attention heads, we introduce two complementary statistics:

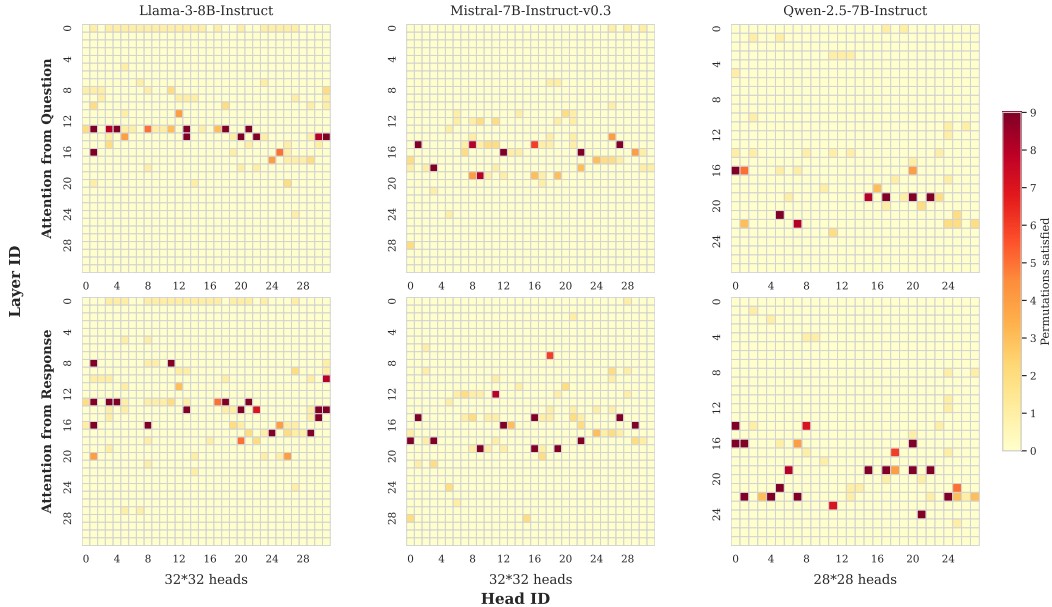

Figure 3: Distribution of Sensitive Heads across layers and head indices. Top: Question-as-Source; Bottom: Response-as-Source. Color intensity reflects the number of permutations in which a head satisfies the Sensitive Head criteria, with darker shades indicating higher consistency. Expert Heads, which satisfy criteria across all permutations, appear in the darkest color.

**Activation Frequency** $f_\pi^{(l,h)}$: measures the proportion of samples in which a head $(l, h)$ is activated under permutation $\pi$, reflecting its consistency:

$$f_\pi^{(l,h)} = \frac{1}{|\mathcal{S}|} \sum_{s \in \mathcal{S}} \text{Activated}(l,h)_{src}^{\pi,s}. \tag{4}$$

Here, $S$ denotes the set of all input samples.

**Average Attention Score** $\bar{A}_\pi^{(l,h)}$: measures the mean attention allocated by head $(l, h)$ to gold documents across activated samples, reflecting its ability to concentrate on task-relevant information:

$$\bar{A}_\pi^{(l,h)} = \frac{\sum_{s \in \mathcal{S}} \text{Activated}(l,h)_{src}^{\pi,s} \cdot \sum_{j=1}^{n} A_{src \to G_j}^{(l,h),s}}{\sum_{s \in \mathcal{S}} \text{Activated}(l,h)_{src}^{\pi,s}}. \tag{5}$$

To systematically evaluate activated heads, we adopt two thresholds: (1) **Activation frequency threshold** $\tau_f = 0.6$, requiring that a head be activated in more than 60% of samples under a given permutation. (2) **Average attention score percentile** $\tau_p = 0.9$, retaining only the top 10% of activated heads ranked by average attention to gold documents. These thresholds strike a balance between reliability (frequent activation) and focus (strong attention to evidence).

**Sensitive and Expert Heads.** An Activated head is classified as a **Sensitive Head** if it satisfies both thresholds for *at least one permutation*:

$$\text{SensitiveHeads} = \left\{ (l,h) \,\middle|\, \exists \pi, \; f_\pi^{(l,h)} > \tau_f \; \wedge \; \bar{A}_\pi^{(l,h)} > P\tau_p\big(\bar{A}_\pi^{(l',h')}\big) \right\}. \tag{6}$$

A Sensitive Head is further promoted to an **Expert Head** if it consistently meets both thresholds across *all permutations*:

$$\text{ExpertHeads} = \left\{ (l,h) \,\middle|\, \forall \pi, \; f_\pi^{(l,h)} > \tau_f \; \wedge \; \bar{A}_\pi^{(l,h)} > P\tau_p\big(\bar{A}_\pi^{(l',h')}\big) \right\}. \tag{7}$$

The full selection procedure is summarized in Alg. 1 in Appendix C.

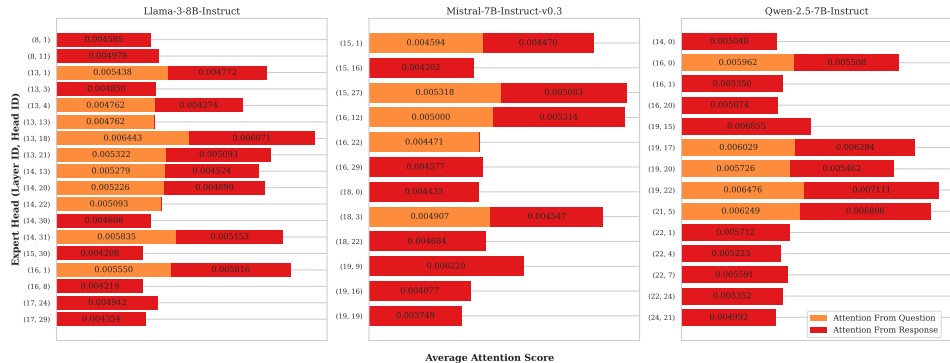

Figure 4: Head-wise distribution of Expert Heads and their average attention scores. Horizontal stacked bars show contributions from Question-as-Source (orange) and Response-as-Source (red). The concentration of Expert Heads differs across models, reflecting distinct layer-specific roles in semantic integration and evidence selection.

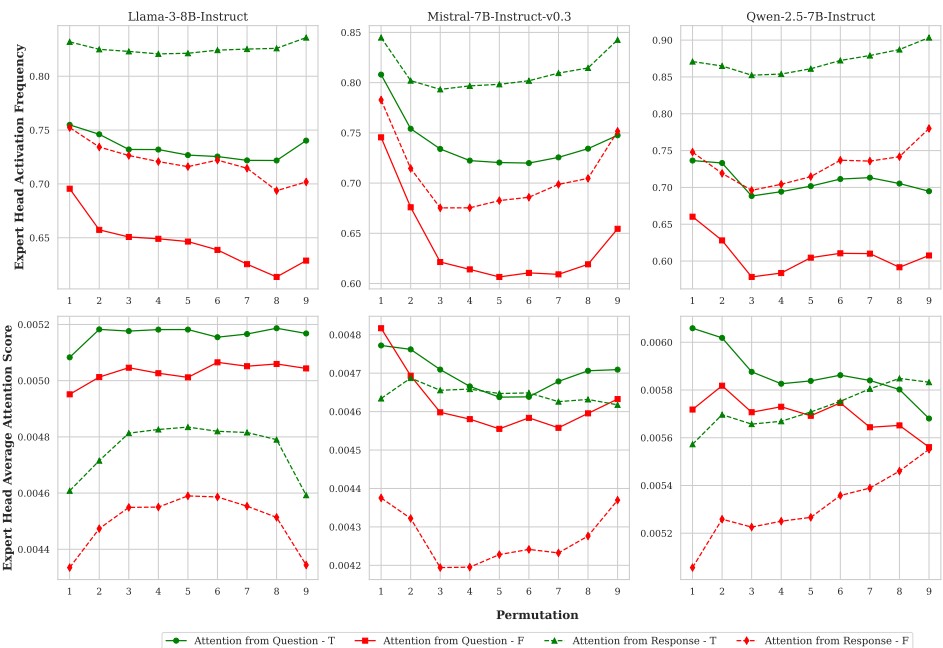

Figure 5: Relationship between Expert Head activation and model answer correctness. Green curves denote correct answers, red curves incorrect ones; solid lines represent Question-as-Source, dashed lines Response-as-Source. Top: activation frequency; Bottom: average attention score. Expert Heads are more active and concentrated in correct cases, while weaker or dispersed activation correlates with errors. The x-axis (1–9) indicates the gold document's position, from beginning (1) to end (9).

## 3.2 EXPERT HEAD ANALYSIS

Figure 3 illustrates the distribution of Sensitive Heads across model layers and head indices. For LLaMA-3-8B-Instruct and Mistral-7B-Instruct-v0.3, Sensitive Heads are concentrated in middle layers, underscoring the role of these layers in semantic integration. In contrast, Qwen-2.5-7B-Instruct exhibits a greater concentration of Sensitive Heads in deeper layers, suggesting that later layers specialize in evidence selection.

Among these Sensitive Heads, those that remain consistently active across all permutations are identified as **Expert Heads**. Figure 4 shows that, under Query-as-Source, Expert Heads form a smaller but more focused subset of the broader pool engaged under Response-as-Source. This indicates that, while answer generation involves a larger set of heads for evidence integration, Expert Heads retain sharper focus on task-relevant documents.

Figure 5 further demonstrates the relationship between Expert Head activation and model answer correctness. When answers are correct, Expert Heads show both higher activation frequencies and stronger average attention scores on gold documents. Conversely, when answers are incorrect, Expert Head activation is weaker and their attention becomes more dispersed, leading to poorer evidence integration and an increased risk of hallucination.

## 4 EVALUATION OF EXPERT HEADS

We evaluate the practical effectiveness of Expert Heads on **document identification and ranking tasks**, rather than directly on QA accuracy. Conventional QA evaluation relies on metrics such as answer correctness or string matching, which may obscure whether a model actually attends to task-relevant evidence. Performance can also be influenced by components beyond attention (e.g., feed-forward layers or decoding strategies), making it difficult to isolate the contribution of evidence integration. In contrast, document identification and ranking provide a more controlled and interpretable setting, allowing us to directly assess whether Expert Heads reliably focus on critical documents.

### 4.1 EXPERIMENTAL SETTING

**Task Definition.** Given a query $Q$, two gold documents $\{G_1, G_2\}$, and eight distractor documents $\{D_1, \ldots, D_8\}$, the model must identify and rank the gold documents among all candidates. We evaluate performance using both identification-oriented metrics (**Precision@2**) and ranking-oriented metrics (**NDCG@2** and **MAP**) to provide a comprehensive assessment.

**Datasets.** Experiments are conducted on the test sets of HotpotQA (2,269 samples), 2WikiMultiHopQA (Ho et al., 2020) (2,471 samples), and MuSiQue (Trivedi et al., 2022) (2,486 samples), providing diverse scenarios for evaluating document identification and ranking.

**Document Ranking with Expert Head Voting.** We evaluate Expert Heads derived from both Query-as-Source and Response-as-Source attention across LLaMA-3-8B-Instruct, Mistral-7B-Instruct-v0.3, and Qwen-2.5-7B-Instruct. For each model and each attention source, we first select Expert Heads based on activation frequency $\tau_f$ and average attention score percentile $\tau_p$, fixing the number of Expert Heads at **five per setting** to ensure fairness.

Given a query and candidate documents, each Expert Head independently produces a ranking based on its attention scores *from the query to each candidate*. Final rankings are then aggregated through a **voting scheme** across the five Expert Heads. Details of the thresholds and voting procedure are provided in Appendix B and C.

**Baselines.** We compare Expert Heads against widely used retrieval models: BM25 (Robertson et al., 2009), DPR (Karpukhin et al., 2020), Contriever (Izacard et al., 2021), MiniLM (Wang et al., 2020), GTR (Ni et al., 2021), ColBERTv2 (Santhanam et al., 2021), BGE (Xiao et al., 2024), Qwen3 (Zhang et al., 2025) and LLM Rank, which refers to direct document ranking by model generation.

### 4.2 MAIN RESULTS

As shown in Table 1, experimental results demonstrate that Expert Heads substantially surpass all baseline retrieval models across all datasets, indicating their effectiveness in document ranking tasks. Importantly, they provide a principled and interpretable mechanism for document ranking. By focusing on a small, consistently activated subset of heads, they highlight the model components most responsible for evidence integration, offering insights into reasoning patterns without relying on black-box full-model outputs.

Table 1: Precision@2 (P@2), NDCG@2, and MAP performance across three datasets. Each base model includes three variants: Expert Heads (Question), Expert Heads (Response), and LLM Rank. Baselines include BM25, DPR, Contriever, MiniLM, GTR, ColBERTv2, BGE, and Qwen3. Best results are shown in **bold**; second-best are underlined.

| Method | HotpotQA | | | 2WikiMultiHopQA | | | MuSiQue | | |
|---|---|---|---|---|---|---|---|---|---|
| | P@2 | NDCG@2 | MAP | P@2 | NDCG@2 | MAP | P@2 | NDCG@2 | MAP |
| BM25 | 57.47 | 50.23 | 60.58 | 52.77 | 45.64 | 56.80 | 49.30 | 43.27 | 54.66 |
| DPR | 60.14 | 52.97 | 63.17 | 61.27 | 54.85 | 64.79 | 58.61 | 50.82 | 61.50 |
| Contriever | 61.50 | 55.19 | 64.90 | 59.51 | 52.15 | 62.07 | 62.37 | 54.04 | 64.01 |
| MiniLM | 71.04 | 64.04 | 71.82 | 73.07 | 65.72 | 73.43 | 65.47 | 56.75 | 66.37 |
| GTR | 64.26 | 57.32 | 66.65 | 70.03 | 61.89 | 70.26 | 64.42 | 55.94 | 65.78 |
| ColBERTv2 | 64.63 | 57.31 | 66.58 | 68.70 | 61.21 | 69.89 | 60.42 | 51.80 | 62.25 |
| BGE | 75.23 | 69.45 | 76.37 | 77.12 | 71.06 | 77.76 | 70.25 | 62.89 | 71.26 |
| Qwen3 | 68.55 | 61.77 | 70.21 | 73.27 | 65.84 | 73.68 | 66.55 | 57.54 | 67.05 |
| **LLaMA-3-8B-Instruct** | | | | | | | | | |
| Expert Heads (Q) | 88.23 | 89.97 | 94.03 | 73.47 | 77.84 | 85.71 | 82.18 | 84.78 | 90.08 |
| Expert Heads (R) | **90.72** | **91.98** | **95.08** | 77.30 | 81.58 | 87.83 | **83.57** | **86.05** | **90.95** |
| LLM Rank | 66.31 | 70.06 | 78.22 | 76.49 | 79.65 | 86.60 | 69.63 | 73.70 | 80.93 |
| **Mistral-7B-Instruct-v0.3** | | | | | | | | | |
| Expert Heads (Q) | 88.08 | 89.73 | 93.60 | 70.72 | 75.27 | 83.21 | 80.03 | 82.98 | 88.76 |
| Expert Heads (R) | 88.89 | 90.50 | 94.13 | 73.27 | 77.62 | 84.83 | 80.53 | 83.40 | 88.85 |
| LLM Rank | 36.45 | 39.97 | 55.03 | 45.14 | 49.82 | 62.22 | 39.72 | 44.31 | 57.54 |
| **Qwen-2.5-7B-Instruct** | | | | | | | | | |
| Expert Heads (Q) | 86.93 | 88.69 | 93.19 | 74.69 | 78.47 | 85.72 | 81.30 | 83.85 | 89.44 |
| Expert Heads (R) | 88.45 | 90.02 | 93.87 | **77.94** | **81.88** | **88.29** | 82.70 | 85.02 | 90.11 |
| LLM Rank | 77.81 | 78.70 | 86.11 | 76.57 | 80.89 | 87.15 | 76.65 | 79.68 | 88.97 |

## 4.3 ABLATION STUDY

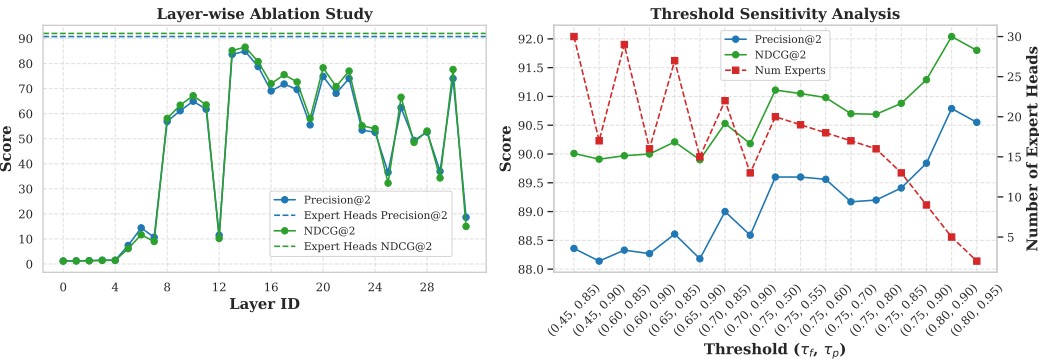

Figure 6: Ablation study results. Left: Layer-wise ablation, where all heads within one layer are treated as Expert Heads. Results show that middle layers contribute most, while lower layers play a limited role and final layers degrade performance. Right: Threshold sensitivity for Response-as-Source Expert Heads across varying activation thresholds. Performance improves under stricter criteria, even though the number of selected heads decreases sharply, indicating that tighter thresholds yield more specialized and effective subsets.

As shown in Fig. 6, we further conduct ablation experiments to analyze layer contributions and threshold sensitivity on LLaMA-3-8B-Instruct using the HotpotQA test set (2,269 samples), with Precision@2 and NDCG@2 as evaluation metrics.

**Layer-wise Ablation.** We evaluate the contribution of different layers by treating all heads from a given layer as expert heads. Results reveal that middle layers contribute the most to performance,

confirming their critical role in semantic integration and evidence selection, while lower layers play a limited role. Interestingly, using the final layer for document identification and ranking leads to a significant drop in performance, possibly because the model is already focused on preparing to generate the next token.

**Threshold Sensitivity.** We further analyze the effect of varying activation thresholds on Response-as-Source expert heads. As thresholds become stricter, the number of selected expert heads decreases substantially. Interestingly, instead of degrading, performance gradually improves under stricter criteria. This indicates that higher thresholds effectively filter out less informative heads, leaving a smaller yet more specialized subset that contributes more strongly to evidence identification and ranking. Thus, Expert Heads not only remain robust to threshold selection but can even benefit from tighter activation constraints.

## 5 DISCUSSIONS

**Expert Heads provide a natural signal for context reranking, pruning, and interpretability.** Because they consistently highlight task-critical evidence, their attention patterns can guide context reranking by moving highly attended documents closer to the end of the context, or context pruning by discarding distractors that receive negligible attention. This head-guided mechanism reduces computational cost in long-context settings while preserving reasoning quality. In addition, Expert Heads make the model's reasoning process more interpretable: instead of black-box ranking scores, users can directly observe which documents are prioritized by the most reliable heads, yielding transparent explanations for model outputs.

**Expert Heads offer a principled approach to hallucination mitigation and factuality detection.** Our experiments show that correct answers are associated with stronger and more concentrated Expert Head activations, while incorrect answers display weaker or dispersed activation. This suggests that activation strength and focus can be used as a diagnostic signal to detect hallucinations in real time. Responses accompanied by weak or diffuse Expert Head patterns can be flagged as unreliable, enabling factuality-aware decoding strategies that enforce tighter grounding of answers to verifiable evidence, particularly in multi-document reasoning scenarios.

**Expert Heads can also serve as guidance for model distillation and reinforcement learning.** In knowledge distillation, the attention maps of Expert Heads in a teacher model can be transferred to a student model, ensuring that evidence-centric reasoning is preserved during compression. In reinforcement learning or Reinforcement Learning from Human Feedback (RLHF), Expert Head activation can be incorporated into the reward function, encouraging models to prioritize task-relevant documents during reasoning. This bridges interpretability and training, providing a way to align model optimization not only with human preferences but also with evidence fidelity, ultimately leading to more reliable and controllable language models.

## 6 RELATED WORK

**Retrieval and Ranking Methods.** Retrieval has evolved from sparse term-based approaches to dense neural representations (Johnson et al., 2019; Khattab & Zaharia, 2020; Khattab et al., 2021a;b; Santhanam et al., 2022) and embeddings derived from large language models (Zhang et al., 2023; Xiao et al., 2023). Modern encoders prioritize efficiency and generalization, while recent open-source models (Li et al., 2023a; Zhang et al., 2024) enhance multilingual capabilities, instruction-following, and long-context handling. In parallel, LLM-based ranking methods directly leverage language models as rerankers (Zhuang et al., 2023; 2024a;b; Drozdov et al., 2023), often achieving superior semantic matching and reasoning, albeit with higher computational cost.

**Functional Specialization and Interpretability of Attention Heads.** Early work (Jain & Wallace, 2019; Clark et al., 2019) questioned whether raw attention weights provide faithful explanations, showing they can be misleading without controlled analysis. Later studies(Kovaleva et al., 2019; Vig, 2019) demonstrated that attention becomes informative when paired with interventions—e.g., head ablation (Michel et al., 2019; Voita et al., 2019), attention-weight manipulation (Serrano & Smith, 2019; Wiegreffe & Pinter, 2019), causal probing (Pruthi et al., 2019), and counterfactual edits (Abnar & Zuidema, 2020)—which can identify heads that causally influence model behav-

ior. More recent research (Li et al., 2023b; Kumar et al., 2024; Zheng et al., 2024) highlights functional specialization of specific heads—for example, induction heads (Ren et al., 2024), name-mover heads (Tigges et al., 2023; García-Carrasco et al., 2024). These findings support treating certain attention heads as mechanistic primitives that drive model behavior, enabling targeted interpretability and modification. Our work extends this line of inquiry by identifying Expert Heads as stable mechanisms for robust evidence integration.

## 7 CONCLUSION

In this work, we identify **Expert Heads**, a small subset of attention heads that consistently focus on task-critical evidence across permutations, models, and attention sources. Our findings reveal layer-specific functional specialization in LLMs: mid-layer heads in LLaMA and Mistral dominate semantic integration, while deeper-layer heads in Qwen specialize in evidence selection. We demonstrate that Expert Heads provide interpretable signals for evidence integration, enabling improvements in document identification, ranking, and hallucination detection. Beyond performance gains, they establish a transparent and stable mechanism that links internal model reasoning with external behavior. Overall, Expert Heads serve both as an interpretability tool and a practical mechanism to enhance efficiency, reliability, and controllability in large language models.

## 8 ACKNOWLEDGMENTS

This work is supported by the National Natural Science Foundation of China (No. 62576236), the Key-Area Research and Development Program of Guangdong Province (2024B0101050005), the Natural Science Foundation of Jiangsu Province (BK20251823), Beijing Natural Science Foundation (4262049), Suzhou Key Laboratory of Artificial Intelligence and Social Governance Technologies (SZS2023007), Smart Social Governance Technology and Innovative Application Platform (YZCXPT2023101), the Innovation System of the Integration between Industry and Education for Smart Governance (CJRH2024101), the Leadership Talent Program (Science and Education) of SIP, the Fundamental Research Funds for the Central Universities, JLU, the Priority Academic Program Development of Jiangsu Higher Education Institutions, the open research fund of Suzhou Key Lab of Multi-modal Data Fusion and Intelligent Healthcare and the Science and Technology Development Plan of Suzhou City (Science and Technology Research Program (Medical and Health Innovation)) (Grant No. SYW2025065).

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

## A    THE USE OF LARGE LANGUAGE MODELS (LLMs)

In preparing this paper, **GPT-5-mini** was employed exclusively for **grammar checking and minor language polishing**. No scientific content, experimental design, or interpretation of results was generated by any LLM. All ideas, analyses, and conclusions presented are solely those of the authors.

## B    IMPLEMENTATION DETAILS

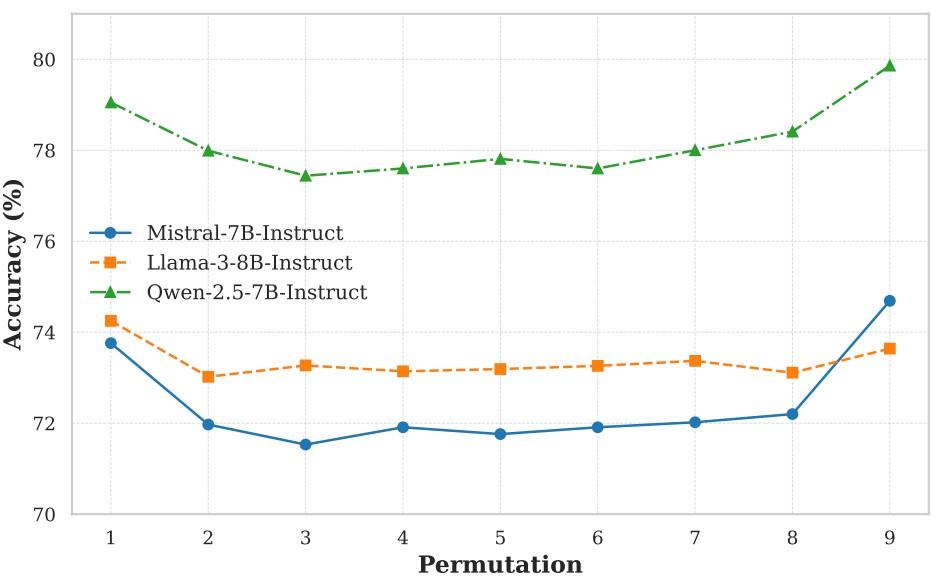

Figure 7: Effect of gold document position on multi-hop QA accuracy. The x-axis (1–9) denotes the gold document's location, from start (1) to end (9). Accuracy peaks when gold documents are placed at the beginning or end of the context, and drops when they appear in the middle—demonstrating strong positional sensitivity in model performance.

Table 2: Thresholds and selected Expert Heads for each model and attention source.

| Model | Attention Source | $(\tau_f, \tau_p)$ | Expert Heads (Layer Id, Head Id) |
|---|---|---|---|
| LLaMA-3-8B-Instruct | Question | (0.65, 0.9) | (13,4), (14,13), (14,20), (14,22), (16,1) |
| LLaMA-3-8B-Instruct | Response | (0.8, 0.9) | (13,18), (14,13), (16,1), (16,8), (17,24) |
| Mistral-7B-Instruct-v0.3 | Question | (0.6, 0.9) | (15,1), (15,27), (16,12), (16,22), (18,3) |
| Mistral-7B-Instruct-v0.3 | Response | (0.75, 0.9) | (15,1), (15,27), (16,12), (18,3), (19,9) |
| Qwen-2.5-7B-Instruct | Question | (0.6, 0.9) | (16,0), (19,17), (19,20), (19,22), (21,5) |
| Qwen-2.5-7B-Instruct | Response | (0.8, 0.93) | (19,15), (19,22), (21,5), (22,1), (22,7) |

Experiments were conducted on 8 NVIDIA V100 GPUs with 32GB memory each. For model responses, the maximum input length was set to 8,192 tokens and the maximum generation length to 4,096 tokens with greedy decoding. All retrieval models used a maximum input length of 512 tokens, while LLM Rank was restricted to a maximum generation length of 30 tokens with greedy decoding. To ensure reproducibility, all random seeds for model inference, data sampling, and dataset splitting were fixed at 42.

For Expert Head performance evaluation, in order to maintain five heads per category, the activation frequency and average attention score percentile thresholds were set as shown in Table 2.

For threshold sensitivity analysis, the number of selected Expert Heads ranged from 1 to 30.

## C  ALGORITHMIC DETAILS

---

**Algorithm 1** Expert Head Selection

---

**Require:** Query $Q$, Gold Documents $\{G_1, \ldots, G_n\}$, Distractor Documents $\{D_1, \ldots, D_m\}$, Attention Tensors $A^{(l,h)}$, Activation Frequency Threshold $\tau_f$, Attention Percentile Threshold $\tau_p$

**Ensure:** Set of Expert Heads $\mathcal{H}_{\text{expert}}$

1: Initialize empty dictionary to store activated heads and their attention scores for all samples and permutations
2: **for** each input sample $s \in \mathcal{S}$ **do**
3:     **for** each input permutation $\pi$ of documents **do**
4:         **for** each attention head $(l, h)$ **do**
5:             **for** each source $src \in \{Q, R\}$ **do**
6:                 Compute attention $A^{(l,h),s}_{src \to G_j}$ and $A^{(l,h),s}_{src \to D_i}$ using Eq. 1 or Eq. 2
7:                 Determine activation using Eq. 3
8:             **end for**
9:             Store activated heads and their attention scores for later aggregation
10:         **end for**
11:     **end for**
12: **end for**
13: **for** each permutation $\pi$ and activated head $(l, h)$ **do**
14:     **for** each source $src \in \{Q, R\}$ **do**
15:         Compute Activation Frequency $f(l, h)^\pi$ using Eq. 4
16:         Compute Average Attention $\bar{A}(l, h)^\pi$ using Eq. 5
17:     **end for**
18: **end for**
19: Identify Sensitive Heads $\mathcal{H}_{\text{sensitive}}$ using Eq. 6
20: Identify Expert Heads $\mathcal{H}_{\text{expert}}$ using Eq. 7
21: **return** $\mathcal{H}_{\text{expert}}$

---

**Algorithm 2** Expert Head Voting for Document Identification and Ranking

---

**Require:** Query $Q$, Candidate Documents $\{C_1, \ldots, C_k\}$, Expert Heads $\mathcal{H}_{\text{expert}}$, Attention Tensors $A^{(l,h)}$, Top-$K$ voting threshold $K$

**Ensure:** Ranked list of documents $\{C_{(1)}, \ldots, C_{(k)}\}$

1: Initialize vote counts: $V(C_i) \leftarrow 0$ for all $i \in [1, k]$
2: **for** each Expert Head $(l, h) \in \mathcal{H}_{\text{expert}}$ **do**
3:     Compute attention scores $S^{(l,h)}(C_i) \leftarrow A^{(l,h)}_{Q \to C_i}$ for all $i$
4:     Rank documents in descending order of $S^{(l,h)}(C_i)$
5:     Select top-$K$ documents from this ranking
6:     **for** each document $C_i$ in top-$K$ **do**
7:         $V(C_i) \leftarrow V(C_i) + 1$                       ▷ This head casts one vote for $C_i$
8:     **end for**
9: **end for**
10: Rank all documents in descending order of $V(C_i)$
11: **return** Ranked document list

---

For clarity and reproducibility, we provide the algorithmic details of the proposed framework in this appendix. Algorithm 1 summarizes the procedure for identifying Expert Heads from raw attention patterns, including input permutations, activation statistics, and consistency checks across different document orderings. Algorithm 2 then illustrates how the selected Expert Heads are leveraged for document identification and ranking, forming the basis of our evaluation experiments in Sec. 4. Together, these algorithms provide a complete framework for Expert Head selection and evaluation.

# D PROMPT TEMPLATES

This appendix presents the prompt templates used in the experiments for multi-hop question answering and document ranking. Templates are tailored to each model family: LLaMA-3-8B-Instruct, Mistral-7B-Instruct-v0.3, and Qwen-2.5-7B-Instruct.

Multi-hop QA Templates: Specify structured document inputs followed by a query. The assistant must generate an answer strictly grounded in the provided documents.

Document Ranking Templates: Present 10 candidate documents and a query. The assistant is instructed to output only a ranked list of document indices, ordered by relevance, without explanations.

These standardized templates ensure consistency across models and tasks, and facilitate reproducibility of our experimental results.

Table 3: Multi-hop Question Answering prompt template for LLaMA-3-8B-Instruct.

| Multi-hop Question Answering for LLaMA-3-8B-Instruct |
|---|
| **Input:** `<|begin_of_text|><|start_header_id|>system<|end_header_id|>` |
| Read the following documents and answer the question based ONLY on the provided information. |
| `<|eot_id|><|start_header_id|>user<|end_header_id|>` |
| |
| Title: · · · |
| Content: · · · |
| |
| Title: · · · |
| Content: · · · |
| |
| · · · |
| |
| Question: {question} |
| `<|eot_id|><|start_header_id|>assistant<|end_header_id|>` |
| |
| **Output:** {response} |

Table 4: Multi-hop Question Answering prompt template for Mistral-7B-Instruct-v0.3.

| Multi-hop Question Answering for Mistral-7B-Instruct-v0.3 |
|---|
| **Input:** `[INST]` |
| Read the following documents and answer the question based ONLY on the provided information. |
| |
| Title: · · · |
| Content: · · · |
| |
| Title: · · · |
| Content: · · · |
| |
| · · · |
| |
| Question: {question} |
| `[/INST]` |
| |
| **Output:** {response} |

Table 5: Multi-hop Question Answering prompt template for Qwen-2.5-7B-Instruct.

| Multi-hop Question Answering for Qwen-2.5-7B-Instruct |
|---|
| **Input:** `<|im_start|>system`
Read the following documents and answer the question based ONLY on the provided information.
`<|im_end|>`

`<|im_start|>user`
Title: $\cdots$
Content: $\cdots$

Title: $\cdots$
Content: $\cdots$

$\cdots$

Question: {question}
`<|im_end|>`

`<|im_start|>assistant`
**Output:** {response} |

Table 6: Document ranking prompt template for LLaMA-3-8B-Instruct.

| Document ranking for LLaMA-3-8B-Instruct |
|---|
| **Input:** `<|begin_of_text|><|start_header_id|>user<|end_header_id|>`
You are an expert document ranker. You are given 10 documents and a question. Rank all documents from most relevant to least relevant in answering the question.

Document [1]:
Title: $\cdots$
Content: $\cdots$

Document [2]:
Title: $\cdots$
Content: $\cdots$

$\cdots$

Question: {question}

Return ONLY a comma-separated list of document numbers, sorted from most relevant to least relevant. Do NOT include any other text or explanation. Example output: 7,3,9,1,5,10,6,2,4,8
`<|eot_id|><|start_header_id|>assistant<|end_header_id|>`

**Output:** {ranking list} |

Table 7: Document ranking prompt template for Mistral-7B-Instruct-v0.3.

| Document ranking for Mistral-7B-Instruct-v0.3 |
|---|
| **Input:** `[INST]`
You are an expert document ranker. You are given 10 documents and a question. Rank all documents from most relevant to least relevant in answering the question.

Document [1]:
Title: · · ·
Content: · · ·

Document [2]:
Title: · · ·
Content: · · ·

. . .

Question: {question}

Return ONLY a comma-separated list of document numbers, sorted from most relevant to least relevant. Do NOT include any other text or explanation. Example output: 7,3,9,1,5,10,6,2,4,8
`[/INST]`

**Output:** {ranking list} |

Table 8: Document ranking prompt template for Qwen-2.5-7B-Instruct.

| Document ranking for Qwen-2.5-7B-Instruct |
|---|
| **Input:** `<|im_start|>system` You are Qwen, created by Alibaba Cloud. You are a helpful assistant. `<|im_end|>`

`<|im_start|>user`
You are an expert document ranker. You are given 10 documents and a question. Rank all documents from most relevant to least relevant in answering the question.

Document [1]:
Title: · · ·
Content: · · ·

Document [2]:
Title: · · ·
Content: · · ·

. . .

Question: {question}

Return ONLY a comma-separated list of document numbers, sorted from most relevant to least relevant. Do NOT include any other text or explanation. Example output: 7,3,9,1,5,10,6,2,4,8
`<|im_end|>`

`<|im_start|>assistant`
**Output:** {ranking list} |

