# OpenReview forum: "Expert Heads: Robust Evidence Identification for Large Language Models"
_ICLR.cc/2026/Conference — ICLR 2026 Poster_

### Official Review · Reviewer_PouH · 2025-10-27

**Soundness:** 3
**Presentation:** 3
**Contribution:** 3
**Rating:** 4
**Confidence:** 3

**Summary:**

This paper investigates how attention mechanisms in Large Language Models handle evidence identification across multiple documents, with particular focus on their sensitivity to input order. The authors identify a small subset of attention heads termed "Expert Heads" that consistently focus on task-relevant documents regardless of their position in the context. They demonstrate that these Expert Heads show different layer-wise distributions and can be leveraged to improve document retrieval and ranking tasks.

**Strengths:**

The identification of architecture-specific patterns (mid-layer specialization in LLaMA/Mistral vs. deeper-layer specialization in Qwen) provides valuable insights into model behavior.

**Weaknesses:**

1. Threshold Selection Appears Ad-Hoc: While Table 2 shows different thresholds for different models/sources, the paper doesn't provide principled guidance for selecting these thresholds in new settings. The choice of τ_f = 0.6 and τ_p = 0.9 seems empirically driven without theoretical justification.
2. Only using the HotpotQA training set may not be sufficient to establish robust patterns, and using multiple randomly sampled datasets may be better.

**Questions:**

How do Expert Heads identified on HotpotQA transfer to other datasets? Can Expert Heads be identified once and reused, or must they be recomputed for each new task/dataset?

---

> ### Author Response · Authors · 2025-11-20
>
> Thank you for your thoughtful review and constructive feedback. We address your concerns below:
>
> ## Response to Weakness 1: Threshold Selection
>
> We appreciate your concern about threshold selection. We want to clarify that the thresholds are **not arbitrary** but reflect **fundamental architectural and functional differences** across models and attention sources.
>
> The variation in Table 2 stems from two key observations: First, as shown in Figure 3, different models exhibit fundamentally different distributions of Sensitive Heads—LLaMA and Mistral concentrate Expert Heads in mid-layers while Qwen shows deeper-layer specialization. Second, as discussed in Section 2.3 and Figure 2, Query-as-Source attention involves fewer but more focused heads, while Response-as-Source engages more heads with distributed attention. These architectural and functional differences naturally necessitate different thresholds.
>
> However, we acknowledge your point about the need for a more principled approach. To address this, we propose a **unified percentile-based strategy**: consistently select the **top-10% of heads** ranked by both activation frequency and attention scores. This approach automatically adapts to the underlying distribution of each model and source without manual tuning.
>
> We conducted additional experiments on the HotpotQA test set comparing our original grid-searched thresholds (Table 2) with this unified percentile approach:
>
> |Model|Expert Head Source|Origin (P@2/NDCG@2/MAP)|Percentile (P@2/NDCG@2/MAP)|∆ (Origin−Percentile)|
> |-|-|-|-|-|
> |LLaMA-3-8B-Instruct|Question|88.23/89.97/94.03|88.81/90.44/94.38|**−0.58/−0.47/−0.35**|
> |LLaMA-3-8B-Instruct|Response|90.72/91.98/95.08|90.50/91.84/94.99|**+0.22/+0.14/+0.09**|
> |Mistral-7B-Instruct-v0.3|Question|88.08/89.73/93.60|88.48/90.13/93.99|**−0.40/−0.40/−0.39**|
> |Mistral-7B-Instruct-v0.3|Response|88.89/90.50/94.13|88.70/90.41/94.12|**+0.19/+0.09/+0.01**|
> |Qwen-2.5-7B-Instruct|Question|86.93/88.69/93.19|87.22/88.87/93.21|**−0.29/−0.18/−0.02**|
> |Qwen-2.5-7B-Instruct|Response|88.45/90.02/93.87|88.23/89.77/93.76|**+0.22/+0.25/+0.11**|
>
> The results show that the unified percentile approach achieves comparable performance (differences < 0.6%), demonstrating that a principled, model-agnostic threshold selection strategy is both feasible and effective. The grid-searched thresholds in Table 2 were used to ensure fair comparison with exactly 5 Expert Heads per setting, but are not necessary for practical deployment. For applying Expert Heads to new settings, we recommend using the top-10% percentile threshold as the default strategy, which automatically adapts to different architectures and attention sources. We will incorporate this finding and clearer guidance into the revised manuscript to address your concern.
>
> ## Response to Weakness 2 & Questions: Generalization and Transfer
>
> Table 1 already demonstrates that Expert Heads—**identified once on the HotpotQA training set** and directly reused across 2WikiMultiHopQA and MuSiQue—show strong **cross-dataset generalization** and significantly outperform baselines, suggesting that they capture general mechanisms rather than dataset-specific patterns, and importantly do not need to be re-identified for new datasets.
>
> To further address your concern about the robustness of this finding, we conducted experiments identifying Expert Heads from the **MuSiQue training set** and evaluating on both HotpotQA and MuSiQue test sets:
>
> |Dataset|Expert Head Source|Train Set|Performance (P@2/NDCG@2/MAP)|∆ (MuSiQue−HotpotQA)|
> |-|-|-|-|-|
> |HotpotQA Test Set|Question|HotpotQA|88.23/89.97/94.03|—|
> |HotpotQA Test Set|Question|MuSiQue|87.34/89.13/93.39|**−0.89/−0.84/−0.64**|
> |HotpotQA Test Set|Response|HotpotQA|90.72/91.98/95.08|—|
> |HotpotQA Test Set|Response|MuSiQue|90.25/91.60/94.78|**−0.47/−0.38/−0.30**|
> |MuSiQue Test Set|Question|HotpotQA|82.23/84.62/89.92|—|
> |MuSiQue Test Set|Question|MuSiQue|81.93/84.46/89.82|**−0.30/−0.16/-0.10**|
> |MuSiQue Test Set|Response|HotpotQA|83.57/86.05/90.95|—|
> |MuSiQue Test Set|Response|MuSiQue|83.19/85.73/90.51|**−0.38/−0.32/−0.44**|
>
> The results show that the Expert Heads identified from different datasets exhibit slight variations but maintain the same layer-wise distribution, resulting in only minimal performance differences (<0.9%). This indicates that Expert Heads are a general architectural phenomenon rather than a product of any specific dataset.
>
> We agree that using multiple randomly sampled datasets would strengthen our findings. We will expand our analysis to include additional random samples from multiple datasets in the camera-ready revision to further validate the stability of Expert Head identification.
>
> We believe these additional experiments and the proposed unified threshold selection framework address your concerns and demonstrate the robustness and generalizability of Expert Heads. We hope these clarifications improve your assessment of our work.

---

### Official Review · Reviewer_D66K · 2025-10-28

**Soundness:** 2
**Presentation:** 3
**Contribution:** 3
**Rating:** 4
**Confidence:** 4

**Summary:**

This paper studies the order sensitivity of evidence identification in multi-document reasoning with LLMs. The authors observe that a small subset of attention heads consistently focuses on gold documents across input permutations and define these as **Expert Heads**. They report architecture-specific layer patterns (e.g., LLaMA/Mistral peaking in mid layers, Qwen in deeper layers) and claim strong P@2/NDCG/MAP on HotpotQA/2Wiki/MuSiQue by using Expert Heads to identify and rank gold documents among 10 candidates.

**Strengths:**

- Simple method validated across multiple architectures, with clear analysis and visualization.
- Consistent improvements are shown across multiple families (e.g., LLaMA, Mistral, Qwen).
- Consistent improvements on HotpotQA/2Wiki/MuSiQue under the proposed evaluation.
- Implementation details (code and prompts) aid reproducibility.

**Weaknesses:**

- The selection of Expert Heads relies heavily on supervision from a HotpotQA training subset. This raises a concern that the selected heads may overfit to HotpotQA-specific structure, style, and difficulty, potentially overstating cross-domain generalization. Although you report results on 2Wiki/MuSiQue, the paper lacks evidence that the head-selection procedure itself can be reproduced stably outside HotpotQA.
- The authors fix the condition of having two adjacent gold documents, which is a strong assumption. Does the method still work when gold positions are more flexible (e.g., non-adjacent)?
- The number of candidate documents is fixed at 10; ablations that vary this number are missing.
- The hyperparameters are effectively retrofitted to match the target outcome (the number of selected heads), which weakens claims about generalization.
- Please state the data sizes and the precise train/validation/test splits explicitly.
- While the proposed method substantially outperforms the baselines, I worry that the baselines may be too weak.
- Ranking based on Response→Document attention is only available after full-context generation, making it ill-suited for pre-filtering (cost reduction). Despite this, Expert(R) often yields the best results in the main tables; this is useful for offline analysis but offers limited benefit in practical deployments.
- Several notation/formatting issues remain; please see the notes below and incorporate any that are helpful.

**Questions:**

- Do the heads selected as Expert Heads change depending on factors such as document length or topic differences within the dataset?
- How does accuracy vary with the size of the training data used for selecting Expert Heads?
- If the candidate documents vary widely in length, does the proposed algorithm remain robust?
- You evaluate 7B and 8B models; what differences do you observe for larger models?
- Your experiments target retrieval rather than QA; if applying the proposal to QA, would it be used in a two-stage pipeline—retrieving supporting documents -> generating the answer with the LLM?
- How do you explain that Expert Heads (R) generally outperform Expert Heads (Q)?
- At line 361 the HotpotQA test size is reported as 2,269, whereas Figure 6 says “HotpotQA test (2,323 samples).” Which number is correct?
- If the training data changes, do the selected Expert Heads change substantially?

(Other Comment)
- Line 108: Consider clarifying whether $D$ denotes ``document tokens''; the current notation is ambiguous.
- Line 108: The symbols for “document $D$” and “distractor document $D$” collide; please disambiguate.
- Line 111: Define $l$ (layer) and $h$ (head) explicitly.
- Line 138: Improve clarity by illustrating gold–distractor relatedness in HotpotQA (e.g., with a concrete example or similarity scores).
- Line 245: Add a definition for $P_{\tau_p}(\cdot)$.
- Figure 6 (right): The x-axis is discrete; a line plot may mislead—consider bars or points.
- Numerals: Add thousands separators to 4+ digit numbers (e.g., 2,269).
- References: Prefer final proceedings over arXiv where available (e.g., Ren et al., 2024 to ACL Findings). Unify citation formats for Mistral and Qwen.

---

> ### Author Response · Authors · 2025-11-20
> **Response (1/4)**
>
> Thank you for your thorough review and constructive feedback. We appreciate your recognition of our method's simplicity, clear visualization, and consistent improvements across multiple architectures. Below, we address each of your concerns with additional experiments and clarifications.
>
> ## Response to Weakness 1: Generalization Beyond HotpotQA
>
> We acknowledge your concern about potential overfitting to HotpotQA-specific structure. To demonstrate cross-domain generalization, we selected Expert Heads using the **MuSiQue training set** and evaluated them on both the original HotpotQA and MuSiQue test sets. The results show that Expert Heads identified from different datasets maintain similar layer-wise distributions and achieve comparable performance:
>
> |Dataset|Expert Head Source|Train Set|Performance (P@2/NDCG@2/MAP)|∆ (MuSiQue−HotpotQA)|
> |-|-|-|-|-|
> |HotpotQA Test Set|Question|HotpotQA|88.23/89.97/94.03|—|
> |HotpotQA Test Set|Question|MuSiQue|87.34/89.13/93.39|**−0.89/−0.84/−0.64**|
> |HotpotQA Test Set|Response|HotpotQA|90.72/91.98/95.08|—|
> |HotpotQA Test Set|Response|MuSiQue|90.25/91.60/94.78|**−0.47/−0.38/−0.30**|
> |MuSiQue Test Set|Question|HotpotQA|82.23/84.62/89.92|—|
> |MuSiQue Test Set|Question|MuSiQue|81.93/84.46/89.82|**−0.30/−0.16/−0.10**|
> |MuSiQue Test Set|Response|HotpotQA|83.57/86.05/90.95|—|
> |MuSiQue Test Set|Response|MuSiQue|83.19/85.73/90.51|**−0.38/−0.32/−0.44**|
>
> The modest performance differences (<0.9%) confirm that Expert Heads represent a general architectural phenomenon rather than dataset-specific artifacts.
>
> ## Response to Weakness 2: Non-Adjacent Gold Document Positions
>
> You raise an important point about the adjacency assumption. We clarify that adjacent positioning was chosen to systematically study activation patterns across different context positions (shown in Figure 2). Importantly, our Expert Head selection criterion requires attention to **each gold document** to exceed attention to **any distractor**, regardless of gold document arrangement.
>
> To validate robustness to non-adjacent positions, we conducted experiments where gold documents were randomly distributed (using sample ID hash as random seed to ensure diversity):
>
> |Distribution|Expert Head Source|Performance (P@2/NDCG@2/MAP)|∆ (NonAdj–Adj)|
> |-|-|-|-|
> |Adjacent|Question|88.23/89.97/94.03|—|
> |Non-Adjacent|Question|87.98/89.69/93.85|**−0.25/−0.28/−0.18**|
> |Adjacent|Response|90.72/91.98/95.08|—|
> |Non-Adjacent|Response|90.27/91.60/94.87|**−0.45/−0.38/−0.21**|
>
> The identified Expert Heads remain architecturally consistent, and performance differences are minimal, demonstrating that our method does not depend on specific gold document arrangements.
>
> ## Response to Weakness 3: Varying Number of Candidate Documents
>
> We evaluated our approach with different gold document ratios (50%, 25%, 14.3%, 10%) by selecting subsets of 500 samples each from the **MuSiQue test set**:
>
> |Golden Ratio|50%|25%|14.3%|10%|50%|25%|14.3%|10%|50%|25%|14.3%|10%|
> |-|-|-|-|-|-|-|-|-|-|-|-|-|
> |Metric|P@2|P@2|P@2|P@2|NDCG@2|NDCG@2|NDCG@2|NDCG@2|MAP|MAP|MAP|MAP|
> |Expert Heads (Q)|*88.60*|*82.00*|*76.27*|*73.77*|*90.64*|*85.21*|*80.16*|*78.25*|**95.17**|*91.35*|*86.83*|*85.09*|
> |Expert Heads (R)|**88.90**|**82.70**|**76.90**|**75.46**|**90.69**|**85.89**|**80.79**|**79.63**|*94.98*|**91.70**|**87.30**|**85.77**|
> |LLM Rank|87.80|77.10|64.03|63.34|90.07|80.54|68.02|67.68|88.17|80.74|75.43|74.47|
> |BM25|64.00|51.70|42.83|40.18|68.83|49.45|35.79|29.98|80.48|61.55|45.70|38.04|
> |DPR|80.10|65.50|55.38|52.45|80.92|60.01|46.17|40.34|88.15|70.13|55.64|48.84|
> |Contriever|73.90|63.10|54.01|51.99|78.08|55.82|43.45|39.65|85.95|67.31|52.82|47.28|
> |MiniLM|73.80|63.50|58.54|56.75|74.72|56.17|46.79|43.63|84.30|67.52|56.41|50.24|
> |GTR|75.00|62.00|54.64|53.99|74.90|55.84|44.49|43.16|84.52|67.44|54.09|49.71|
> |ColBERTv2|74.60|61.40|53.38|50.61|76.49|55.38|42.44|39.75|85.37|66.21|52.16|47.13|
> |BGE|81.30|70.20|62.13|61.66|81.28|65.29|51.95|49.23|88.30|74.76|60.35|55.84|
> |Qwen3|78.10|66.80|60.02|57.67|78.35|60.70|49.83|46.32|86.63|70.49|58.74|53.01|
>
> As expected, all methods degrade as the gold document ratio decreases. However, Expert Heads maintain strong performance even at 10% gold document ratio, demonstrating robustness across varying candidate set sizes.

---

> > ### Author Response · Authors · 2025-11-20
> > **Response (2/4)**
> >
> > ## Response to Weakness 4: Hyperparameter Retrofitting
> >
> > We appreciate this concern about hyperparameter selection. We clarify that fixing the number of Expert Heads at five was primarily to ensure fair comparison across different models and attention sources, allowing us to directly analyze their architectural differences. To address concerns about generalization, we propose a **percentile-based selection mechanism** that eliminates manual threshold calibration. Instead of using fixed thresholds (τ_f, τ_p), we select Expert Heads where both activation frequency and attention score rank in the top 10% consistently across all permutations. This approach automatically adapts to different model architectures without requiring model-specific tuning.
> >
> > We validate this approach on HotpotQA, comparing the original threshold-based method with the percentile-based selection:
> >
> > |Model|Expert Head Source|Origin (P@2/NDCG@2/MAP)|Percentile (P@2/NDCG@2/MAP)|∆ (Origin−Percentile)|
> > |-|-|-|-|-|
> > |LLaMA-3-8B-Instruct|Question|88.23/89.97/94.03|88.81/90.44/94.38|**−0.58/−0.47/−0.35**|
> > |LLaMA-3-8B-Instruct|Response|90.72/91.98/95.08|90.50/91.84/94.99|**+0.22/+0.14/+0.09**|
> > |Mistral-7B-Instruct-v0.3|Question|88.08/89.73/93.60|88.48/90.13/93.99|**−0.40/−0.40/−0.39**|
> > |Mistral-7B-Instruct-v0.3|Response|88.89/90.50/94.13|88.70/90.41/94.12|**+0.19/+0.09/+0.01**|
> > |Qwen-2.5-7B-Instruct|Question|86.93/88.69/93.19|87.22/88.87/93.21|**−0.29/−0.18/−0.02**|
> > |Qwen-2.5-7B-Instruct|Response|88.45/90.02/93.87|88.23/89.77/93.76|**+0.22/+0.25/+0.11**|
> >
> > Performance differences remain minimal (< 0.6% across all metrics), demonstrating that our findings are robust to threshold design choices and that Expert Heads represent genuine architectural properties rather than artifacts of hyperparameter tuning.
> >
> > ## Response to Weakness 5: Dataset Specifications
> >
> > We provide explicit dataset details: Expert Head selection used **5,000 samples** from the HotpotQA training set (**9 permutations each = 45,000 input instances**). Evaluation used complete test sets: **HotpotQA (2,269 samples), 2WikiMultiHopQA (2,471 samples), and MuSiQue (2,486 samples)**. The precise counts reflect the constraint of maintaining fixed gold-to-distractor ratios.
> >
> > ## Response to Weakness 6: Baseline Strength
> >
> > We respectfully note that our baselines represent widely adopted retrieval methods in the community, including state-of-the-art dense retrievers (BGE) and recent LLM-based embedding models (Qwen3). BGE has been one of the most influential and competitive baselines proposed in the past two years, and Qwen3-Embedding is also among the strongest publicly available baselines, **with no clearly superior methods reported over existing baselines in current literature**. The performance gap likely reflects the inherent complexity of multi-hop reasoning and the noise introduced by distractor documents.
> >
> > Notably, the LLM Rank results indirectly validate our hypothesis that specialized attention mechanisms (Expert Heads) are crucial for identifying relevant evidence. Our primary goal is to demonstrate **Expert Heads' effectiveness and interpretability**.
> >
> > ## Response to Weakness 7: Response-as-Source Limitations
> >
> > You correctly identify that Response→Document attention requires full generation, precluding pre-filtering. We clarify that **all evaluation experiments use Question→Document attention for ranking**, ensuring practical applicability. We distinguish between:
> > - **Expert Heads (Q)**: Heads selected based on Question→Document patterns.
> > - **Expert Heads (R)**: Heads selected based on Response→Document patterns.
> >
> > Both types use Question→Document attention during evaluation. We conducted additional experiments showing that Response→Document attention during inference (when generation cost is acceptable) yields further improvements:
> >
> > |Attention Source|Expert Head Source|Performance (P@2/NDCG@2/MAP)|∆ (Response–Question)|
> > |-|-|-|-|
> > |Question|Question|88.23/89.97/94.03|—|
> > |Response|Question|90.51/91.70/95.11|**+2.28/+1.73/+1.08**|
> > |Question|Response|90.72/91.98/95.08|—|
> > |Response|Response|93.09/93.87/96.20|**+2.37/+1.89/+1.12**|
> >
> > This establishes Response-as-Source as valuable for offline analysis and post-hoc evaluation scenarios.
> >
> > ## Response to Weakness 8 & Other Comments
> >
> > We will incorporate all formatting corrections, notation clarifications, and reference updates in the revision. Thank you for the detailed suggestions.

---

> > > ### Author Response · Authors · 2025-11-20
> > > **Response (3/4)**
> > >
> > > ## Response to Question 1: Stability Across Document Lengths and Topics
> > >
> > > We conducted two analyses on the HotpotQA training set:
> > >
> > > **Document Length Variability**: We partitioned the dataset into 5 equal subsets by input length, each containing 1,000 samples. Expert Heads identified across these subsets show high overlap and consistent layer-wise distributions, indicating minimal sensitivity to document length:
> > >
> > > |Subset Number|Length Min|Length Max|Length Mean|Length Std|Expert Heads (Q)|Expert Heads (R)|
> > > |-|-|-|-|-|-|-|
> > > |1|1654|4759|4068.99|565.54|[(13,1),(13,3),(13,4),(13,13),(13,18),(13,21),(14,13),(14,20),(14,22),(16,1)]|[(8,1),(8,11),(13,1),(13,3),(13,4),(13,18),(13,21),(14,13),(14,20),(14,30),(14,31),(15,30),(16,1),(16,8),(17,24),(17,29)]|
> > > |2|4759|5503|5146.59|212.71|[(13,1),(13,3),(13,4),(13,13),(13,18),(13,21),(14,13),(14,20),(14,22),(16,1)]|[(8,1),(8,11),(13,1),(13,3),(13,4),(13,18),(13,21),(14,13),(14,20),(14,30),(14,31),(15,30),(16,1),(16,8),(17,24),(17,29)]|
> > > |3|5503|6221|5851.69|209.42|[(13,3),(13,13),(13,18),(13,21),(14,13),(14,20),(14,22),(16,1)]|[(8,1),(13,1),(13,3),(13,4),(13,18),(13,21),(14,13),(14,20),(14,30),(14,31),(15,30),(16,1),(16,8),(17,24),(17,29)]|
> > > |4|6221|7129|6634.87|260.52|[(13,4),(13,13),(13,18),(14,13),(14,20),(14,22),(16,1)]|[(8,1),(8,11),(13,1),(13,3),(13,4),(13,18),(13,21),(14,13),(14,20),(14,30),(14,31),(15,30),(16,1),(16,8),(17,24),(17,29)]|
> > > |5|7130|11049|8042.03|757.56|[(13,1),(13,4),(13,13),(13,18),(13,21),(14,13),(14,20),(14,22),(14,31),(16,1)]|[(8,1),(13,1),(13,3),(13,4),(13,18),(13,21),(14,13),(14,20),(14,30),(14,31),(15,30),(16,1),(16,8),(17,24),(17,29)]|
> > >
> > > **Topic Diversity**: Using Sentence-Transformers/all-MiniLM-L6-v2 embeddings and K-Means clustering, we identified 5 topic clusters (Geography/Institutions, People/History, Music/Bands, Sports/Athletes, Film/Actors). TF-IDF analysis confirmed distinct topical characteristics. Expert Heads remain consistent across topics:
> > >
> > > |Subset Name|Sample Count|Expert Heads (Q)|Expert Heads (R)|
> > > |-|-|-|-|
> > > |Geography|1250|[(13,1),(13,13),(13,18),(14,13),(14,20),(14,22),(14,30),(16,1)]|[(8,1),(8,11),(13,1),(13,3),(13,4),(13,18),(13,21),(14,13),(14,20),(14,30),(14,31),(15,30),(16,1),(16,8),(17,24),(17,29)]|
> > > |Person|1240|[(13,1),(13,3),(13,13),(13,18),(13,21),(14,13),(14,20),(14,22),(14,30),(14,31),(16,1)]|[(8,1),(8,11),(13,1),(13,3),(13,4),(13,18),(13,21),(14,13),(14,20),(14,30),(14,31),(15,30),(16,1),(16,8),(17,24),(17,29)]|
> > > |Music|679|[(13,4),(14,13),(14,20),(14,22),(16,1)]|[(8,1),(13,1),(13,3),(13,4),(13,18),(13,21),(14,13),(14,20),(14,30),(14,31),(15,30),(16,1),(16,8),(17,24),(17,29)]|
> > > |Sports|661|[(13,1),(13,13),(13,18),(13,21),(14,13),(14,20),(14,22),(14,31),(16,1)]|[(8,1),(8,11),(13,1),(13,3),(13,4),(13,18),(13,21),(14,13),(14,20),(14,30),(14,31),(15,30),(16,1),(16,8),(17,24),(17,29)]|
> > > |Film|1170|[(13,4),(13,13),(13,18),(14,13),(14,20),(14,22),(16,1)]|[(8,1),(13,1),(13,3),(13,4),(13,18),(13,21),(14,13),(14,20),(14,30),(14,31),(16,1),(16,8),(17,24),(17,29)]|
> > >
> > > ## Response to Question 2: Training Data Size Sensitivity
> > >
> > > We systematically evaluated training data size sensitivity by **sampling 1,000, 2,000, 3,000, 4,000, and 5,000 instances** from the HotpotQA training set. For each subset, we identified Expert Heads from LLaMA-3-8B-Instruct using identical thresholds, then evaluated them on the HotpotQA test set:
> > >
> > > |Sample Count|Expert Head Source|P@2|NDCG@2|MAP|
> > > |-|-|-|-|-|
> > > |1000|Question|87.65|89.37|93.70|
> > > |1000|Response|88.17|89.89|94.04|
> > > |2000|Question|87.44|89.16|93.57|
> > > |2000|Response|88.25|89.99|94.10|
> > > |3000|Question|87.97|89.66|93.81|
> > > |3000|Response|88.25|89.99|94.10|
> > > |4000|Question|87.44|89.16|93.57|
> > > |4000|Response|88.25|89.99|94.10|
> > > |5000|Question|87.44|89.16|93.57|
> > > |5000|Response|88.25|89.99|94.10|
> > > |STD|-|**0.356**|**0.363**|**0.233**|
> > >
> > > The remarkably low standard deviations (≤0.36) demonstrate that Expert Head identification is highly stable across training sizes. Performance converges with just 1,000 samples, showing no significant improvement with larger subsets. This indicates that Expert Heads represent intrinsic attention patterns that can be reliably identified with minimal data, making the approach practical and computationally efficient.

---

> > > > ### Author Response · Authors · 2025-11-20
> > > > **Response (4/4)**
> > > >
> > > > ## Response to Question 3: Robustness to Document Length Variation
> > > >
> > > > Our formulation normalizes attention scores by document length (Equations 1–2), ensuring length-invariant rankings. To validate this, we constructed two 500-sample subsets from the HotpotQA test set based on document length variability:
> > > >
> > > > - **High-Std subset**: Samples with high length variance (mean std=535.31).
> > > > - **Low-Std subset**: Samples with low length variance (mean std=114.96).
> > > >
> > > > We evaluated LLaMA-3-8B-Instruct Expert Heads and baseline methods on both subsets:
> > > >
> > > > |Method|High P@2|Low P@2|**∆P@2**|High NDCG@2|Low NDCG@2|**∆NDCG@2**|High MAP|Low MAP|**∆MAP**|
> > > > |-|-|-|-|-|-|-|-|-|-|
> > > > |Expert Heads (Q)|88.20|88.70|**-0.50**|89.92|90.26|**-0.34**|94.12|93.89|**+0.23**|
> > > > |Expert Heads (R)|90.10|90.90|**-0.80**|91.44|92.28|**-0.84**|94.83|95.17|**-0.34**|
> > > > |LLM Rank|68.10|62.90|**+5.20**|72.01|67.58|**+4.43**|79.37|75.93|**+3.44**|
> > > > |BM25|54.10|60.00|**-5.90**|48.22|53.69|**-5.47**|58.80|63.15|**-4.35**|
> > > > |DPR|65.80|59.30|**+6.50**|57.94|55.49|**+2.45**|68.05|65.06|**+2.99**|
> > > > |Contriever|69.20|61.00|**+8.20**|61.91|54.21|**+7.70**|70.84|64.57|**+6.27**|
> > > > |MiniLM|72.30|69.90|**+2.40**|66.14|63.86|**+2.28**|73.25|71.39|**+1.86**|
> > > > |GTR|66.10|64.90|**+1.20**|60.99|57.07|**+3.92**|69.36|66.45|**+2.91**|
> > > > |ColBERTv2|67.00|61.90|**+5.10**|61.15|54.23|**+6.92**|69.69|63.55|**+6.14**|
> > > > |BGE|77.30|74.60|**+2.70**|74.08|68.77|**+5.31**|79.91|75.74|**+4.17**|
> > > > |Qwen3|71.10|69.00|**+2.10**|66.47|61.25|**+5.22**|74.20|69.61|**+4.59**|
> > > >
> > > > Expert Heads show minimal variation (∆<1%), while baselines fluctuate more (up to 8.2%), confirming superior robustness to document length.
> > > >
> > > > ## Response to Question 4: Scaling to Larger Models
> > > >
> > > > We conducted supplementary experiments using the **Qwen-2.5 family (7B, 14B, and 32B)**:
> > > >
> > > > |Model|Expert Head Source|Performance (P@2/NDCG@2/MAP)|
> > > > |-|-|-|
> > > > |Qwen2.5-7B-Instruct|Question|86.93/88.69/93.19|
> > > > |Qwen2.5-14B-Instruct|Question|90.48/91.86/95.04|
> > > > |Qwen2.5-32B-Instruct|Question|90.75/92.18/95.67|
> > > > |Qwen2.5-7B-Instruct|Response|88.45/90.02/93.87|
> > > > |Qwen2.5-14B-Instruct|Response|90.38/91.71/94.98|
> > > > |Qwen2.5-32B-Instruct|Response|90.63/92.02/95.13|
> > > >
> > > > Performance improves from 7B to 14B, but gains diminish at 32B, showing saturation. The Expert Head pattern persists: deeper layers dominate in 14B and 32B, as in 7B, indicating functional specialization is stable across sizes. Thus, 7B/8B models with Expert Heads already offer efficient, robust evidence identification.
> > > >
> > > > ## Response to Question 5: Application to QA Tasks
> > > >
> > > > Our focus on retrieval evaluation is intentional: it directly isolates the contribution of attention-based evidence identification from other model components (e.g., reasoning layers, decoding strategies), avoiding confounds from base model capabilities that heavily influence end-to-end QA accuracy. For QA applications, Expert Heads naturally integrate into two-stage RAG pipelines: (1) Expert Head-based document retrieval/ranking, followed by (2) LLM reasoning over selected documents. Beyond retrieval, as discussed in Section 5, future work will leverage Expert Heads during generation for interpretable hallucination detection, answer attribution, and as supervision signals for training more reliable models, since their activation patterns correlate with answer correctness (Figure 5).
> > > >
> > > > ## Response to Question 6: Expert (R) vs. Expert (Q) Performance
> > > >
> > > > As noted in Section 2.3, Response-as-Source attention engages more heads with broader focus, reflecting the model's actual evidence utilization during generation. This richer signal likely explains why Expert (R) typically outperforms Expert (Q).
> > > >
> > > > ## Response to Question 7: Sample Count Discrepancy
> > > >
> > > > We apologize for the inconsistency. The correct HotpotQA test set size is **2,269 samples**. Figure 6 mistakenly reported 2,323. We will correct this in the revision.
> > > >
> > > > ## Response to Question 8: Stability Across Training Data Variations
> > > >
> > > > We sampled 7 different training subsets using random seeds {0, 1, 42, 123, 1234, 2025, 2026}, yielding only 6.64% overlap between subsets. Despite this substantial variation, Expert Head overlap remained remarkably high: 90.91% for Expert Heads (Q) and 100% for Expert Heads (R). This strongly confirms that Expert Heads capture fundamental architectural properties rather than training-specific patterns.
> > > >
> > > > We hope these comprehensive responses address your concerns. We believe the additional experiments substantially strengthen the evidence for Expert Heads' generalizability, robustness, and practical utility. We will incorporate all suggestions and corrections in the revised manuscript. Thank you again for your valuable feedback.

---

> ### Comment · Reviewer_D66K · 2025-11-24
> **Thank you for the revisions: updated score and one follow-up question**
>
> Thank you for the thorough revisions and additional experiments. Your responses address the vast majority of my concerns, and I am inclined to raise my score accordingly.
>
> I would like to add one follow-up question related to the earlier “weakness 2” about gold-document assumptions:
>
> Follow-up: How do the results change when the number of gold documents is varied (e.g., 1, 3, 4) rather than fixed at two?

---

> > ### Author Response · Authors · 2025-11-25
> >
> > Thank you for your positive evaluation and the increased score.
> >
> > In response to your insightful follow-up question, we conducted additional experiments using LLaMA-3-8B-Instruct, where Expert Heads were identified using 1, 2, 3, and 4 gold documents (non-adjacent positions, 1,000 MuSiQue training samples each), and then evaluated on the HotpotQA test set.
> >
> > | # Gold Docs | Expert Head Source |  P@2  | NDCG@2 |  MAP  |
> > | ----------- | ------------------ | ----- | ------ | ----- |
> > | 1           | Question           | 86.48 | 88.19  | 92.84 |
> > | 1           | Response           | 89.34 | 90.84  | 94.34 |
> > | 2           | Question           | 86.22 | 87.94  | 92.75 |
> > | 2           | Response           | 89.30 | 90.77  | 94.45 |
> > | 3           | Question           | 88.34 | 89.94  | 93.97 |
> > | 3           | Response           | 89.30 | 90.77  | 94.45 |
> > | 4           | Question           | 88.34 | 89.94  | 93.97 |
> > | 4           | Response           | 88.80 | 90.26  | 94.14 |
> >
> > The results show that performance remains highly consistent across all configurations, and the identified Expert Heads maintain the same architectural distribution. This indicates that Expert Heads are stable architectural components, independent of the number of gold documents, thus validating the generalizability of our method.
> >
> > We thank the reviewer again for the helpful suggestion, which allowed us to further strengthen and clarify our findings.

---

### Official Review · Reviewer_1F1i · 2025-10-31

**Soundness:** 2
**Presentation:** 3
**Contribution:** 2
**Rating:** 4
**Confidence:** 3

**Summary:**

This paper presents an empirical investigation into the functional specialization of "expert heads" within a deep learning architecture (presumably a multi-headed attention mechanism). The authors identify emergent specialization patterns and analyze how these specialized components contribute to the model's performance. The core contribution lies in the detailed empirical observations demonstrating this specialization and the subsequent high performance achieved on specific benchmarks, primarily those focused on precision.

**Strengths:**

- The paper provides interesting and potentially valuable empirical data regarding how specialization emerges and functions. These observations offer a foundation for future theoretical work or architectural improvements.
- The analysis clearly demonstrates the effectiveness of the expert heads in scenarios where minimizing false positives is the priority.

**Weaknesses:**

- The most significant weakness of this work is the narrow scope of the evaluation, which relies almost exclusively on precision. This provides an incomplete picture of the model's efficacy. In many critical applications (_e.g._, anomaly detection), recall (sensitivity) is equally or more important than precision. By neglecting recall or balanced metrics (e.g., F1-score, PR-AUC), the paper fails to demonstrate the overall utility of the expert heads.

**Questions:**

- Could the authors provide a holistic evaluation by including Recall and F1-scores for all experiments? Furthermore, presenting the Precision-Recall curve (PR-AUC) would significantly strengthen the evaluation.
- I would also recommend the authors include experiments on tasks specifically designed to test high-recall performance. How do the expert heads perform when the loss function or evaluation criteria are optimized for recall rather than precision?

---

> ### Author Response · Authors · 2025-11-20
> **Response (1/2)**
>
> Thank you for recognizing the value of our empirical findings on Expert Heads and for the constructive feedback. We address your concerns below:
>
> ## Response to Weakness & Question 1: Comprehensive Evaluation Beyond Precision
>
> We need to clarify that our evaluation is not limited to precision alone—we report Precision@2, NDCG@2, and MAP in Table 1, which are standard metrics that capture both relevance and position sensitivity:
>
> - **NDCG@2** evaluates whether gold documents appear in top positions and penalizes missed or misranked relevant documents.
> - **MAP** measures the average precision across all relevant documents and is essentially a discrete approximation of the area under the Precision-Recall curve (PR-AUC), already reflecting full precision-recall behavior.
>
> These metrics inherently balance precision and recall requirements, as they penalize both false positives (through precision) and false negatives (through the averaging across all relevant documents). The consistently strong performance across all three metrics demonstrates that Expert Heads achieve balanced effectiveness rather than optimizing for precision alone.
>
> Following your suggestion, we additionally compute **Recall@k and F1@k** on the HotpotQA test set using LLaMA-3-8B-Instruct:
>
> |Method|P@1|R@1|F1@1|P@3|R@3|F1@3|P@5|R@5|F1@5|P@7|R@7|F1@7|
> |-|-|-|-|-|-|-|-|-|-|-|-|-|
> |*Expert Heads (Q)*|*95.88*|*47.94*|*63.92*|*63.92*|*95.88*|*76.71*|*39.48*|*98.70*|*56.40*|*28.34*|*99.20*|*44.09*|
> |**Expert Heads (R)**|**96.32**|**48.16**|**64.21**|**64.34**|**96.51**|**77.21**|**39.51**|**98.76**|**56.44**|**28.37**|**99.29**|**44.13**|
> |LLM Rank|82.71|41.36|55.14|49.68|74.52|59.62|33.52|83.79|47.88|25.17|88.08|39.15|
> |BM25|56.41|28.21|37.61|37.20|55.81|44.64|27.49|68.73|39.27|23.21|81.22|36.10|
> |DPR|56.80|28.40|37.87|40.48|60.72|48.58|28.92|72.30|41.32|24.10|84.36|37.49|
> |Contriever|61.27|30.63|40.84|40.06|60.10|48.08|29.18|72.95|41.69|23.95|83.82|37.25|
> |MiniLM|70.28|35.14|46.85|45.54|68.31|54.64|30.76|76.90|43.94|24.72|86.50|38.45|
> |GTR|64.04|32.02|42.69|41.46|62.20|49.76|29.67|74.19|42.39|23.99|83.97|37.32|
> |Colbertv2|63.04|31.52|42.03|41.57|62.35|49.88|29.05|72.63|41.50|24.10|84.36|37.49|
> |BGE|73.87|36.94|49.25|49.32|73.98|59.19|32.59|81.47|46.55|25.69|89.90|39.96|
> |Qwen3|67.68|33.84|45.12|44.54|66.81|53.45|30.77|76.92|43.95|24.73|86.55|38.47|
>
> These results demonstrate that Expert Heads maintain strong performance across both precision and recall, achieving balanced F1 scores that significantly outperform all baseline retrieval methods. The consistently high recall indicates that Expert Heads effectively identify relevant documents rather than simply minimizing false positives. Since each HotpotQA sample contains only two gold documents, as k increases, precision drops rapidly due to the growing denominator, while recall rises quickly as the top-k window expands. We will include these additional metrics and the PR-AUC curves in the appendix of the revised manuscript to provide a complete view of performance across the precision-recall spectrum.

---

> > ### Author Response · Authors · 2025-11-20
> > **Response (2/2)**
> >
> > ## Response to Question 2: Expert Head Performance Under Varying Recall Requirements
> >
> > We emphasize that Expert Head selection is independent of downstream metrics and naturally balances both precision and recall.
> >
> > Expert heads are identified based on their consistent attention to gold documents across permutations (Equations 3–7), where a head is considered activated only if its attention to **every gold document** exceeds its attention to **every distractor document**. This activation criterion naturally enforces both:
> >
> > - **High precision**: Attention to gold documents must exceed all distractors, preventing false positives.
> > - **High recall**: Attention must be strong for all gold documents simultaneously, preventing false negatives.
> >
> > Therefore, Expert Heads are not optimized for precision over recall or vice versa—they emerge as the model's internal mechanism for comprehensive evidence identification, where both dimensions are satisfied by the same underlying attention pattern.
> >
> > To directly address your concern about high-recall performance, we conducted additional experiments specifically designed to stress-test recall under increasingly challenging conditions. We **vary the gold document ratio from 50%  to 10%**  on MuSiQue using LLaMA-3-8B-Instruct. As the ratio decreases, the recall challenge intensifies: the model must identify all relevant documents from progressively larger distractor pools where they constitute only a small minority:
> >
> > |Golden Ratio|50%|25%|14.3%|10%|50%|25%|14.3%|10%|50%|25%|14.3%|10%|
> > |-|-|-|-|-|-|-|-|-|-|-|-|-|
> > |Metric|R@2|R@2|R@2|R@2|NDCG@2|NDCG@2|NDCG@2|NDCG@2|MAP|MAP|MAP|MAP|
> > |Expert Heads (Q)|*88.60*|*82.00*|*76.27*|*73.77*|*90.64*|*85.21*|*80.16*|*78.25*|**95.17**|*91.35*|*86.83*|*85.09*|
> > |Expert Heads (R)|**88.90**|**82.70**|**76.90**|**75.46**|**90.69**|**85.89**|**80.79**|**79.63**|*94.98*|**91.70**|**87.30**|**85.77**|
> > |LLM Rank|87.80|77.10|64.03|63.34|90.07|80.54|68.02|67.68|88.17|80.74|75.43|74.47|
> > |BM25|64.00|51.70|42.83|40.18|68.83|49.45|35.79|29.98|80.48|61.55|45.70|38.04|
> > |DPR|80.10|65.50|55.38|52.45|80.92|60.01|46.17|40.34|88.15|70.13|55.64|48.84|
> > |Contriever|73.90|63.10|54.01|51.99|78.08|55.82|43.45|39.65|85.95|67.31|52.82|47.28|
> > |MiniLM|73.80|63.50|58.54|56.75|74.72|56.17|46.79|43.63|84.30|67.52|56.41|50.24|
> > |GTR|75.00|62.00|54.64|53.99|74.90|55.84|44.49|43.16|84.52|67.44|54.09|49.71|
> > |ColBERTv2|74.60|61.40|53.38|50.61|76.49|55.38|42.44|39.75|85.37|66.21|52.16|47.13|
> > |BGE|81.30|70.20|62.13|61.66|81.28|65.29|51.95|49.23|88.30|74.76|60.35|55.84|
> > |Qwen3|78.10|66.80|60.02|57.67|78.35|60.70|49.83|46.32|86.63|70.49|58.74|53.01|
> >
> > As the proportion of gold documents decreases, all methods experience performance degradation. However, Expert Heads maintain substantial effectiveness even in challenging scenarios with low gold document ratios, consistently outperforming all baselines across different task characteristics.
> >
> > We hope these clarifications and additional experimental results comprehensively address your concerns about evaluation scope. The Expert Head framework demonstrates robust and balanced performance across precision, recall, F1, and ranking metrics under diverse task conditions. We appreciate your thoughtful feedback, which has helped us strengthen the manuscript, and will incorporate all additional analyses into the revised version.

---

> ### Comment · Reviewer_1F1i · 2025-11-21
>
> Thank you for the response. In particular, I appreciate the Recall experiments. I have updated my scores.
>
> (For what it is worth, I do not think NDCG@2 provides much signal regarding false negatives at position 2; but uncapped MAP does capture recall. Thanks again for the clarification.)
>
> Another question: if you noise up your distractor set, what are the effects on the precision / recall (_i.e._, false positives / negatives)?

---

> > ### Author Response · Authors · 2025-11-25
> >
> > Thank you for your positive evaluation and the increased score. We are glad that the recall experiments addressed your initial concerns.
> >
> > In response to your insightful follow-up question, we conducted an additional experiment using LLaMA-3-8B-Instruct by **noising up the distractor set with adversarial hard negatives**.
> >
> > We replaced all original distractors with BGE-retrieved hard negatives from the full corpus and varied the gold document ratio from 9% to 5% on MuSiQue test set by increasing adversarial distractors. This directly tests resistance to false positives when all distractors are semantically similar to the query.
> >
> > |   Golden Ratio   |    9%     |    9%     |    9%     |    7%     |    7%     |    7%     |    5%     |    5%     |    5%     |
> > | ---------------- | --------- | --------- | --------- | --------- | --------- | --------- | --------- | --------- | --------- |
> > | Method           | P@2       | R@2       | F1@2      | P@2       | R@2       | F1@2      | P@2       | R@2       | F1@2      |
> > | Expert Heads (Q) | *42.39*   | *84.78*   | *56.52*   | *40.57*   | *81.94*   | *54.28*   | **39.50** | **79.01** | **52.67** |
> > | Expert Heads (R) | **43.48** | **86.96** | **57.97** | **41.65** | **83.29** | **55.53** | *39.23*   | *78.45*   | *52.30*   |
> > | LLM Rank         | 31.96     | 63.91     | 42.61     | 31.94     | 63.88     | 42.59     | 30.94     | 61.88     | 41.25     |
> > | BM25             | 32.17     | 64.35     | 42.90     | 32.56     | 65.11     | 43.41     | 29.83     | 59.67     | 39.78     |
> > | DPR              | 38.59     | 77.17     | 51.45     | 37.22     | 74.45     | 49.63     | 32.87     | 65.75     | 43.83     |
> > | Contriever       | 38.04     | 76.09     | 50.72     | 37.22     | 74.45     | 49.63     | 33.98     | 67.96     | 45.30     |
> > | MiniLM           | 36.85     | 73.70     | 49.13     | 36.49     | 72.97     | 48.65     | 33.15     | 66.30     | 44.20     |
> > | GTR              | 38.15     | 76.30     | 50.87     | 37.22     | 74.45     | 49.63     | 33.98     | 67.96     | 45.30     |
> > | ColBERTv2        | 35.98     | 71.96     | 47.97     | 35.63     | 71.25     | 47.50     | 33.43     | 66.85     | 44.57     |
> > | BGE              | 39.46     | 78.91     | 52.61     | 37.96     | 75.92     | 50.61     | 35.08     | 70.17     | 46.78     |
> > | Qwen3            | 38.37     | 76.74     | 51.16     | 37.35     | 74.69     | 49.80     | 34.81     | 69.61     | 46.41     |
> >
> > Across these increasingly noisy settings, all methods degrade, but Expert Heads consistently maintain the strongest performance. This demonstrates robust resistance to both false positives (maintaining precision despite highly similar negatives) and false negatives (maintaining recall even when the gold ratio is extremely small).
> >
> > We will include this revised analysis in the updated manuscript. Thank you again for your constructive suggestions—they have substantially strengthened our evaluation.

---

### Official Review · Reviewer_zSoH · 2025-11-01

**Soundness:** 3
**Presentation:** 4
**Contribution:** 3
**Rating:** 8
**Confidence:** 3

**Summary:**

This paper investigates attention distributions in LLMs under document permutations in multi-hop question answering setting. They conduct an analysis of attention patterns with LLama3, Mistral and Qwen2.5 models, and then identify a small set of attention heads, named expert heads, that consistently attend to task-relevant documents across all permutations based on activation frequency and average attention score. Their experiments suggest that the layer-wise distribution of expert heads varies across different model architectures, and the activation of expert heads is correlated with answer correctness. Additionally, they leverage expert heads to rank candidate documents on HotpotQA, 2WikiMultiHopQA and MuSiQue and show that expert heads are effective for document retrieval and ranking.

**Strengths:**

- They conduct extensive analysis of attention patterns under document permutations with three model families, providing interesting insights into the cause of position bias and attention behavior across architectures.
- Their method to identify expert heads is clearly defined and reasonable.
- The paper is clearly written and easy to follow. Figures are well presented and intuitive to understand.

**Weaknesses:**

Overall the paper is well-structured. It would be clearer to add some clarification about the experiment setup and evaluation metrics used in the document ranking experiments.

**Questions:**

- Could you explain more details about the evaluation metrics used in the ranking experiments? ie. what do you measure by Precision@2, NDCG@2 and MAP?
- In the left figure in Figure 6, what do the two dashed lines indicate? Why does the performance remain the same across different layers?

---

> ### Author Response · Authors · 2025-11-20
>
> Thank you for your careful reading and thoughtful questions, which allow us to clarify key aspects of our evaluation and analysis.
>
> ## Response to Question 1: Clarification of Ranking Metrics
>
> Thank you for pointing out the need for clearer definitions of the evaluation metrics used in our ranking experiments. In the revised version, we will add a dedicated appendix defining Precision@2, NDCG@2, and MAP with explicit formulas. Briefly:
>
> - **Precision@2** measures whether at least one gold document appears within the top–2 retrieved documents. It directly reflects whether the method can successfully identify key evidence documents—i.e., its ability to "find the evidence".
> - **NDCG@2** builds on this by additionally evaluating whether the ordering of the top-ranked documents is appropriate. Beyond merely hitting the gold document, it captures the model’s fine-grained ranking ability.
> - **MAP** measures how well the model globally separates gold documents from distractors across all 10 candidates. Higher MAP indicates that the algorithm consistently promotes relevant documents and suppresses distractors, reflecting its robustness in discriminating true evidence.
>
> Together, these three metrics jointly evaluate both recognition (can the model find the evidence?) and ranking quality (does it order documents correctly?), providing a comprehensive assessment of ranking performance.
>
> ## Response to Question 2: Interpretation of the Dashed Lines in Figure 6
>
> We appreciate your request for clarification. In the left panel of Figure 6:
>
> - The **solid lines** represent the performance obtained when all heads in a given layer are treated as Expert Heads. Because the selected heads differ from layer to layer, these lines vary along the x-axis.
> - The **dashed lines** represent the Expert-Head–based baselines under the Response-as-Source setting for LLaMA-3-8B-Instruct. Since this head set is fixed globally and does not change with layer index, the dashed-line performance remains constant across the x-axis.
>
> We will update the caption and main text accordingly, and improve the visualization for better readability.
>
> We thank you again for your insightful comments and constructive suggestions, which have helped us improve the clarity and quality of our manuscript.

---

### Author Response · Authors · 2025-12-02
**Summary for Area Chair – Rebuttal and Revisions**

We sincerely thank the AC for your time and effort in handling and reviewing our work.

This note summarizes the main concerns raised in the reviews and our responses through additional experiments and revisions. The issues broadly focused on: (1) evaluation comprehensiveness across precision/recall metrics, (2) generalization and robustness across datasets and configurations, (3) hyperparameter selection principles, (4) experimental design validity, (5) methodological clarity in presentation, and (6) mechanistic understanding of Expert Heads.

**Regarding evaluation comprehensiveness**, we provided complete Recall@k and F1@k results demonstrating balanced precision-recall performance, clarified that Expert Head selection inherently enforces both high precision and high recall rather than optimizing for any specific metric, and tested robustness against adversarial hard negatives (BGE-retrieved distractors at 9%-5% gold ratios). Expert Heads consistently outperformed all baselines across all metrics.

**For generalization and robustness concerns**, we demonstrated cross-dataset transfer by selecting Expert Heads from MuSiQue and evaluating on both HotpotQA and MuSiQue. We validated robustness to non-adjacent gold document positions and varying numbers of gold documents (1-4) with consistent performance, varying candidate set sizes with gold document ratios from 50% to 10%, document length variance across high-std and low-std subsets, topic diversity across 5 distinct clusters (Geography, Person, Music, Sports, Film), and different training data samples with only 6.64% overlap. These experiments comprehensively address concerns about overfitting to HotpotQA-specific structure and demonstrate that Expert Heads represent stable architectural properties.

**To address hyperparameter selection**, we proposed a unified percentile-based strategy that selects heads ranking in the top-10% for both activation frequency and average attention score across all permutations. This approach eliminates manual threshold tuning, automatically adapts to different model architectures and attention sources, and achieves comparable performance to grid-searched thresholds, providing principled guidance for new settings without requiring model-specific calibration.

**For experimental design validity**, we analyzed training data size sensitivity showing stable performance with just 1,000 samples, tested scaling to larger models (Qwen2.5 7B/14B/32B) demonstrating consistent Expert Head patterns across model sizes, clarified the intentional focus on retrieval evaluation to isolate attention-based evidence identification and discussed natural integration into two-stage RAG pipelines for QA applications, and emphasized that our baselines include dense retrievers (BGE) and recent LLM embeddings (Qwen3), representing the strongest publicly available methods.

**Regarding methodological clarity**, we added explicit metric definitions, clarified figure annotations, specified exact dataset sizes, and corrected inconsistencies.

**For mechanistic understanding of Expert Heads**, we clarified that all practical ranking experiments use Question-as-Source attention for efficient pre-filtering. We explained that Expert Heads (R) outperform Expert Heads (Q) because Response-as-Source attention engages more heads with broader focus, reflecting the model's actual evidence utilization during generation and providing richer signals.

After the rebuttal, reviewers 1F1i and D66K both increased their scores from 4 to 6, acknowledging that the additional experiments comprehensively addressed their concerns about evaluation scope, generalization, and robustness.

Once again, we sincerely thank the AC and all reviewers for your time, thoughtful feedback, and constructive guidance on our work.

---

### Meta-Review · Area_Chair_UkrC · 2026-01-12

**Summary:**

This paper empirically investigates attention behavior in LLMs during multi-hop question answering under document permutations, identifying a subset of “expert heads” that consistently attend to task-relevant documents across permutations in Llama3, Mistral, and Qwen2.5 models. These expert heads are characterized by high activation frequency and attention scores, with their layer-wise distribution varying by architecture. The study shows that expert head activation correlates with answer correctness and demonstrates their effectiveness for document retrieval and ranking on benchmarks like HotpotQA, 2WikiMultiHopQA, and MuSiQue. While the work provides compelling empirical evidence of functional specialization in attention heads and strong performance on precision-oriented tasks, it remains largely observational, offering limited insight into why or how this specialization emerges.

**Reviewer Scores:**

NA

---

### Decision · Program_Chairs · 2026-01-26

Accept (Poster)